# When a Lump Is Not a Cyst: A Case of Superficial Venous Aneurysm of the Hand Diagnosed with High-Resolution Ultrasound

**DOI:** 10.3390/diagnostics15121546

**Published:** 2025-06-17

**Authors:** Antonio Corvino, Orlando Catalano, Corrado Tagliati, Giulio Cocco, Domenico Tafuri, Fabio Corvino, Marco Fogante, Oriana Simonetti

**Affiliations:** 1Medical, Movement and Wellbeing Sciences Department, University of Naples “Parthenope”, 80133 Naples, Italy; an.cor@hotmail.it (A.C.); domenico.tafuri@uniparthenope.it (D.T.); 2Radiology Unit, Istituto Diagnostico Varelli, 80126 Naples, Italy; orlando.catalano@istitutovarelli.it; 3AST Ancona, Ospedale di Comunità Maria Montessori di Chiaravalle, Via Fratelli Rosselli 176, 60033 Chiaravalle, Italy; 4Department of Neuroscience, Imaging and Clinical Sciences, University “G. d’Annunzio”, 66100 Chieti, Italy; cocco.giulio@gmail.com; 5Vascular and Interventional Radiology Department, Cardarelli Hospital, 80131 Naples, Italy; effecorvino@gmail.com; 6Maternal-Child, Senological, Cardiological Radiology and Outpatient Ultrasound, Department of Radiological Sciences, University Hospital of Marche, 60126 Ancona, Italy; marco.fogante@ospedaliriuniti.marche.it; 7Department of Clinical and Molecular Sciences, Dermatology Clinic, Polytechnic Marche University, Via Conca 71, 60126 Ancona, Italy; o.simonetti@staff.univpm.it

**Keywords:** venous, aneurysm, hand, superficial vein, high-frequency ultrasound

## Abstract

Superficial venous aneurysms of the upper extremities are a rare clinical entity, often underdiagnosed and misinterpreted as other soft tissue masses. We present the case of a 28-year-old male patient with a subcutaneous mass on the dorsum of the left hand, diagnosed as a superficial venous aneurysm by high-resolution ultrasound using a probe bandwidth of up to 18 MHz, unchanged at three-month Doppler-ultrasound examination. This case highlights the fundamental role of high-frequency ultrasound in the differential diagnosis and conservative management of such lesions.

**Figure 1 diagnostics-15-01546-f001:**
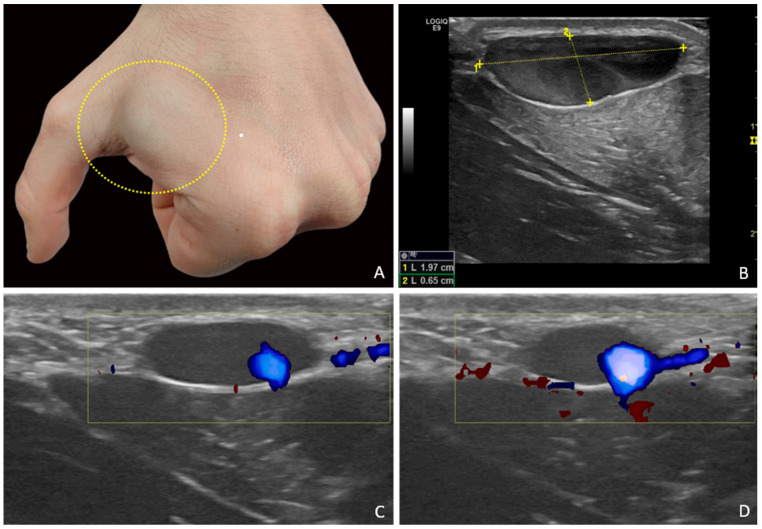
A 28-year-old male patient presented for evaluation of a subcutaneous mass located on the dorsum of the left hand. The mass had been known to the patient for several years and was described as slowly increasing in size over time. It was never associated with pain, signs of functional impairment, or signs of inflammation. Medical history was negative for local trauma or prior interventions in the affected area. A rounded, superficial swelling was observed in the dorso-radial distal area of the left hand, between the first and second metacarpals. The lesion, soft in consistency, compressible on palpation, and non-pulsatile, showed a slight bluish discoloration, which was more evident during fist-clenching (**A**). US examination was performed using a GE LOGIQ™ E9 system with linear probes 9L-D (bandwidth 3.0–8.0 MHz), ML6-15-D (4.5–15.0 MHz), and “hockey stick” L8-18i-D (4.0–15.0 MHz), with and without the interposition of a gel stand-off pad to enhance ultrasound transmission and optimize Doppler signal detection in superficial areas. Technically, the hand of the patient was placed over the medical examination table. The probe was positioned perpendicular to the skin and the lesion width, transverse and longitudinal axes, and distance from skin surface were evaluated. The focus of the color Doppler was located immediately below the lesion. The color Doppler settings were optimized to detect very slow flows, as a low pulse repetition frequency and low wall filter selection are recommended for dermatologic examinations [1]. High-frequency B-mode imaging of the hand lesion obtained using the “hockey stick” probe revealed a well-circumscribed, hypo-anechoic, subcutaneous lesion with thin walls and no calcifications. Distance from skin surface, antero-posterior axis, width, and length were, respectively, 1.5 mm, 6.5 mm, 13 mm, and 20 mm. The lesion was compressible with probe pressure, and contained fine internal mobile echoes, compatible with a “smoke-like” effect (**B**). It was in continuity with a superficial carpo-metacarpal vein from the first-second interdigit space, tributary of the cephalic vein system (Appendix A). Color and power Doppler analysis obtained using the same high-frequency “hockey stick” probe showed very slow venous flow, visible only in a limited eccentric portion of the lesion and only after squeezing maneuvers of the cephalic vein upstream, confirming its functional communication with the superficial venous system (**C**,**D**). No endoluminal thrombi, arteriovenous fistulas, or abnormal communications were noted. The diagnosis was consistent with a superficial VA. A vascular surgeon visit and a Doppler-ultrasound examination after three months were recommended. Conservative management was suggested by the vascular surgeon. The lesion was unchanged at three-month Doppler–ultrasound examination. This paper presents the case of a young patient with a superficial VA of the hand, manifesting as a painless and slowly progressive subcutaneous mass, diagnosed with diagnostic accuracy by high-resolution ultrasound (US). This method allowed for both precise morphologic evaluation and functional characterization of the lesion through a dynamic assessment of venous flow. Despite an extensive literature review, this appears to be the only reported case of a peripheral superficial venous aneurysm documented using high-frequency US probes. In view of the absence of symptoms or complications, the superficial location, and the benign US appearance, a strategy of periodic clinical and US monitoring was adopted. Venous aneurysms (VAs) are defined as focal and segmental dilatations of a venous vessel not associated with varices, arteriovenous communications, or pseudoaneurysms [2]. They are most commonly located in the deep veins of the lower limbs and the cervical region [3]. Involvement of the superficial venous system of the upper extremities is considerably rarer, and the dorsal aspect of the hand represents an exceptional presentation [4]. This rarity contributes to diagnostic difficulty and frequent misinterpretation as other soft tissue masses. High-resolution US plays an important role in identifying these lesions, offering detailed insights into their morphology, vascular continuity, and hemodynamic characteristics [5]. Superficial VAs are rare benign vascular lesions [6]. Diagnosis is often delayed due to their nonspecific clinical presentation and overall low incidence. Their etiology may be congenital, due to alterations in the structural components of the venous wall, or acquired as a result of trauma, local inflammation, or chronic venous hypertension [7]. In this case, the absence of predisposing factors and its slow development over time support the hypothesis of a primary form. Histologically, a primary venous aneurysm is defined as a localized, solitary venous dilatation containing all three histological layers of the vessel wall and connected to a main vein via a single lumen, with no evidence of arteriovenous communication or pseudoaneurysm [2]. Primary VAs have been reported in most major veins. In a study by Gillespie et al., 77% of VAs were located in the lower extremities (with 57% involving the deep venous system); 10% were in the upper limbs; and 13% involved the internal jugular vein. Superficial VAs of the upper limb, and particularly of the hand, are exceedingly rare, with only a few documented cases in the international literature [3,4]. Clinically, VAs typically appear as soft, non-pulsatile subcutaneous nodules that are often asymptomatic. These features can easily lead to misdiagnosis as lipomas, ganglion cysts, or other benign soft tissue masses. Differential diagnosis should also include vascular malformations, isolated varices, and superficial thrombophlebitis [8]. In the literature, the misinterpretation of VAs as synovial cysts is not uncommon [9]. Wang et al. described two cases of wrist VAs initially treated with aspiration as it was assumed that they were ganglion cysts before they were later identified as dilated venous structures [10]. US is a valuable diagnostic tool for the initial evaluation of these lesions due to its non-invasive nature, accessibility, low cost, and ability to provide real-time imaging of superficial structures and vascular flow [5]. Key sonographic features of VAs include a well-defined hypo- or anechoic cavity, continuity with a venous vessel, partial or complete compressibility, absence of thrombi, and slow venous flow, often observed only after dynamic maneuvers [11]. The presence of fine internal moving echoes (“smoke” effect), a finding widely reported in the echocardiography literature, suggests blood stasis without organized thrombus. US also allows for follow-up monitoring of aneurysm size, wall changes, and possible thrombosis, making it a safe and clinically meaningful surveillance tool [12]. In this case, the use of high-frequency linear probes (12 MHz and above) enabled an exceptionally detailed view of the aneurysmal wall and the thin carpo-metacarpal vein, a tributary of the cephalic vein, confirming its anatomical continuity with the aneurysmal sac. These probes, typically used in musculoskeletal and dermatologic imaging, proved highly effective in characterizing superficial vascular lesions even in peripheral locations, enhancing the ability to distinguish vascular masses from other soft tissue lesions [13]. To date, no similar cases have been documented in the literature using such high-frequency US probes for the evaluation of a peripheral superficial VA. An additional technical adjustment used during the US examination was the application of a gel stand-off pad. As described in the literature, this device improves beam focusing, reduces near-field artifacts, and enhances the detection of superficial low-flow vessels, thus improving diagnostic sensitivity in cutaneous and subcutaneous areas [1]. However, it should be considered that computed tomography (CT), magnetic resonance imaging (MRI), and venography are highly informative imaging methods to help confirm the diagnosis and exclude soft-tissue tumors, which typically require further investigation through fine-needle aspiration biopsy (FNAB) to rule out malignancy [8]. In addition, they offer a detailed and comprehensive assessment of the vascular anatomy, enabling a precise evaluation of the venous system, a useful element for surgeons during preoperative planning. Nonetheless, in addition to cost and accessibility limitations, these imaging modalities also have certain drawbacks, including the need for contrast agents and exposure to ionizing radiation (in the case of CT and venography), which is why they are typically reserved for second-line techniques, after US, in the evaluation of superficial lesions [8]. Among the differential diagnoses, venous malformation represents the most challenging entity to distinguish, given its high prevalence among vascular malformations. In fact, cutaneous venous malformations can be clinically indistinguishable from VAs, as they are both slightly bluish masses that are compressible on palpation. However, at US, venous malformation usually shows a heterogeneous architecture, with tubular and curvy anechoic vascular spaces, and possibly acoustically shadowed phleboliths [14]. The main complications associated with VAs include local thrombosis and pulmonary embolism. Although the occurrence of thromboembolic events is considered rare, it is documented in the scientific literature. According to Gabrielli et al., the overall incidence of pulmonary embolism in patients with VAs is between 24% and 32%, mainly involving deep locations. On the contrary, in the case of superficial VAs such as those of cephalic vein, the thromboembolic risk is extremely low, with very few published cases. However, these rare clinical cases demonstrate that these lesions, although considered low risk, should not be underestimated. This justifies a cautious diagnostic and therapeutic approach [15]. However, therapeutic approaches remain highly individualized, as no universally accepted management guidelines are currently available. Asymptomatic, small, superficial VAs may be safely monitored with clinical and US follow-up, without the need for invasive treatment [6,15]. In similar cases described by Li et al. [6] and McKesey et al. [13], conservative management proved safe and effective, supporting the approach adopted in our patient. Surgery is typically indicated only in the presence of pain, thrombosis, rapid growth, thromboembolic risk, or for esthetic or functional reasons [10]. For example, in the cases reported by Lee et al. and Wang et al., treatment consisted of surgical excision and ligation, even in the absence of complications [4,10]. However, surgical decisions are often driven by cosmetic concerns or diagnostic uncertainty. In well-characterized, stable, and asymptomatic lesions, as in our patient, conservative management is fully justified and supported by the current literature [6]. The limitations of this case report are the absence of histopathological analysis, as conservative management was undertaken, and the relatively short follow-up. However, our case contributes to the growing knowledge of this underrecognized condition, emphasizing the importance of accurate diagnosis and differential diagnosis from other soft tissue lesions to avoid unnecessary treatment and ensure a risk-proportional management strategy. In conclusion, superficial VA of the hand is a rare but clinically relevant condition due to its ability to mimic other benign subcutaneous lesions. US, complemented by Doppler techniques, is essential for diagnosis and for guiding the appropriate management strategy. Overall, in the absence of symptoms or complications, a conservative follow-up strategy is both appropriate and evidence-based. This example also demonstrates the utility of high-frequency probes (12 MHz and above) in evaluating superficial vascular conditions, encouraging their broader use in daily US practice.

## Data Availability

Data are contained within the article.

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
