# Peer review of "When a Lump Is Not a Cyst: A Case of Superficial Venous Aneurysm of the Hand Diagnosed with High-Resolution Ultrasound"

_diagnostics, 2025, doi:10.3390/diagnostics15121546_

Round 1
Reviewer 1 Report
Comments and Suggestions for Authors
- Was the lesion diagnosed with only ultrasound? It is not possible in many cases to exclude AVM or venous malformations without CT or MRI. Therefore, the main concern of this report is that the diagnosis was not proven exactly.
- The histopathologic findings were emphasized correctly; however, at least an FNAB should be performed, and the content of the lesion should be shown.
Author Response
Thank you very much for your comments and questions.
- Was the lesion diagnosed with only ultrasound? It is not possible in many cases to exclude AVM or venous malformations without CT or MRI. Therefore, the main concern of this report is that the diagnosis was not proven exactly.
The diagnosis was made using B-mode ultrasound and Doppler ultrasound. All the findings allowed to reach the diagnosis. We thought that it was not useful to perform other imaging examinations.
- The histopathologic findings were emphasized correctly; however, at least an FNAB should be performed, and the content of the lesion should be shown.
It is very dangerous to insert a needle into a vascular lesion, it can cause a serious bleeding. As the diagnosis had already been confidently established non-invasively in this case, such a procedure was considered unnecessary.
Reviewer 2 Report
Comments and Suggestions for Authors
thank you for the invitation to review this manuscript. The authors describe a single patient with a slowly enlarging dorsal-hand mass ultimately identified as a superficial venous aneurysm (VA) by means of high-frequency ultrasound (US). They emphasise the diagnostic value of >12 MHz probes and advocate conservative follow-up. The case is uncommon and the images are instructive; however, several reporting details require attention before acceptance.
Title & Abstract
- Title is catchy but does not state “case report/interesting image”; indexing services may miss the study type.
- Add patient demographics, probe bandwidth (up to 18 MHz) and follow-up strategy in the abstract.
Introduction
- Generally adequate background on VA rarity and misdiagnosis. Would benefit from a short paragraph on potential complications (thrombosis, embolism) to justify imaging urgency.
Methods
- Ultrasound equipment and probes are listed precisely. However, patient positioning, transducer orientation, machine settings (gain, Doppler PRF), distance from skin are missing!
- The manuscript does not state the length of follow-up??; clarify monitoring interval and last clinical assessment.
Results
- Provide precise measurement axes (length × AP × width)
- Include a scale bar in Figure 2
Discussion
- I enjoyed reading the discussion, good job! just mention potential progression to thrombosis/embolism and criteria that would trigger surgery.
Conclusion
• Remove “evidence-based” (overstated for a single case).
References
- During reference checking I noted three specific formatting problems that need correction:
- Reference number 6, journal title should be abbreviated! year is followed by a comma, not a semicolon & volume should include the issue in parentheses.
- Reference number 14, delete the duplicated year please. also use abbreivated journal title. Remove the extraneous closing parenthesis before the period
Author Response
Thank you very much for your comments and suggestions.
Title & Abstract
-Title is catchy but does not state “case report/interesting image”; indexing services may miss the study type.
To improve indexing and adherence to journal standards, we propose revising the title as follows: “When a Lump Isn’t a Cyst: A Case of Superficial Venous Aneurysm of the Hand Diagnosed with High-Resolution Ultrasound”.
We trust this change will enhance the visibility and correct categorization of the article.
-Add patient demographics, probe bandwidth (up to 18 MHz) and follow-up strategy in the abstract
We added patient demographics, probe bandwidth (up to 18 MHz) and follow-up strategy in the abstract.
Introduction
-Generally adequate background on VA rarity and misdiagnosis. Would benefit from a short paragraph on potential complications (thrombosis, embolism) to justify imaging urgency.
We added a paragraph about potential complications (thrombosis, embolism) in the discussion.
Methods
-Ultrasound equipment and probes are listed precisely. However, patient positioning, transducer orientation, machine settings (gain, Doppler PRF), distance from skin are missing!
We added the sentence: “Technically, the hand of the patient was placed over the medical examination table. The probe was positioned perpendicular to the skin and the lesion width, transverse, and longitudinal axes and distance from skin surface were evaluated. The focus of the color Doppler was located immediately below the lesion. The color Doppler settings were optimized to detect very slow flows, as low pulse repetition frequency and low wall filter selection are recommended for dermatologic examinations [5]”.
-The manuscript does not state the length of follow-up??; clarify monitoring interval and last clinical assessment.
A vascular surgeon visit and a Doppler-ultrasound examination after three months were recommended.
Results
-Provide precise measurement axes (length × AP × width)
We provided precise measurement axes (length × AP × width).
-Include a scale bar in Figure 2
We included a scale bar in Figure 2
Discussion
-I enjoyed reading the discussion, good job! just mention potential progression to thrombosis/embolism and criteria that would trigger surgery.
We added in the discussion these sentences: “The main complications associated with venous aneurysms include local thrombosis and pulmonary embolism. Although the occurrence of thromboembolic events is considered rare, it is well documented in the scientific literature. According to Gabrielli et al., the overall incidence of pulmonary embolism in patients with venous aneurysms is between 24% and 32%, mainly involving deep locations. On the contrary, in the case of superficial venous aneurysms, such as those of cephalic vein, the thromboembolic risk is extremely low, with very few published cases. However, these rare clinical cases demonstrate that these lesions, although considered at low risk, should not be underestimated. This justifies a cautious diagnostic and therapeutic approach, especially in symptomatic patients or in the presence of large aneurysms, potentially more prone to blood stasis and thrombus formation. In fact, total excision of a superficial vein system aneurysm can be performed, particularly when the VA is big or associated with pulmonary embolism [14].
References
-During reference checking I noted three specific formatting problems that need correction:
1.Reference number 6, journal title should be abbreviated! year is followed by a comma, not a semicolon & volume should include the issue in parentheses
We modified reference 6 as requested.
2.Reference number 14, delete the duplicated year please. also use abbreviated journal title. Remove the extraneous closing parenthesis before the period
The journal title is reported well; pubmed cite this article with this title; there isn’t an abbreviated form. In the citation 2012 is repeated. If you would like to check this is the link Management of symptomatic venous aneurysm - PubMed Gabrielli R, Rosati MS, Siani A, Irace L. Management of symptomatic venous aneurysm. ScientificWorldJournal. 2012;2012:386478. doi: 10.1100/2012/386478.
Reviewer 3 Report
Comments and Suggestions for Authors
Thank for the opportunity to review this interesting case. The manuscript presents a rare case of a superficial venous aneurysm of the hand, correctly diagnosed using high-resolution ultrasound. I find this report clinically valuable and educational.
The manuscript is well-structured and the methodology is adequately detailed, particularly the ultrasound protocol, which is of high educational interest.
Herein are some minor suggestions
- Discussion: While the literature review is adequate, it could benefit from slightly deeper discussion of differential diagnoses (e.g., venous malformations vs VAs).
- Were there any changes on follow-up?
- Could the authors include any limitation of their case report?
Author Response
Thank you very much for your suggestions and questions.
- Discussion: While the literature review is adequate, it could benefit from slightly deeper discussion of differential diagnoses (e.g., venous malformations vs VAs).
We added a slightly deeper discussion of differential diagnosis, such as venous malformations. These sentences were added at the end of the discussion: “A lesion that needs to be differentiated is venous malformation, which is the most common type of congenital vascular malformation. In fact, cutaneous venous malformation can be clinically indistinguishable from venous aneurism, as they are both slightly bluish masses that are compressible on palpation. However, at ultrasound examination venous malformation usually shows heterogeneous architecture, with tubular and curvy anechoic vascular spaces, and possible acoustically shadowed phleboliths”.
- Were there any changes on follow-up?
No, there were no changes on follow-up.
- Could the authors include any limitation of their case report?
We added some limitations at the end of the discussion. In particular, this sentence was written:” Limitations of this case report are the absence of histopathological analysis as conservative management was performed and the relatively short follow-up”.
Round 2
Reviewer 1 Report
Comments and Suggestions for Authors
Unfortunately, my previous comments were not taken into account. Literature exists regarding fine needle aspiration biopsy (FNAB) in vascular lesions. Additionally, CT and MRI findings are crucial for ruling out malignancies.
Author Response
Thank you very much for your comments and suggestions.
Wwe sincerely apologize if our previous response gave the impression that we had not taken your valuable suggestions into account, this was due to a misunderstanding, and we hope you will kindly accept our apologies. This time, we have tried to be more thorough and to address each of your comments carefully. We truly appreciate your constructive feedback.
In the revised Discussion section, we have specified that “High-resolution US plays an important role in identifying these lesions, offering detailed insights into morphology, vascular continuity, and hemodynamic characteristics,” and that “US is a valuable diagnostic tool for the initial evaluation of these lesions due to its non-invasive nature, accessibility, low cost, and ability to provide real-time imaging of superficial structures and vascular flow.” These revisions aim to avoid conveying the misleading message that ultrasound is the most useful or the only imaging modality to be employed in such cases.
We have removed the sentence “The presence of all these elements may obviate the need for additional imaging such as Computed Tomography or Magnetic Resonance Imaging” from the Discussion section.
Finally, we have added an entire paragraph incorporating your suggestions and enriched it with additional bibliographic references (ref 14, Dubois et al.): “However, it should be considered that computed tomography (CT), magnetic resonance imaging (MRI), and venography are highly informative imaging methods, helping confirm the diagnosis and exclude soft-tissue tumors, which typically require further investigation through fine-needle aspiration biopsy (FNAB) to rule out malignancy. In addition, they offer a detailed and comprehensive assessment of the vascular anatomy, enabling precise evaluation of the venous system, a useful element for surgeons during preoperative planning. Nonetheless, in addition to cost and accessibility limitations, these imaging modalities also have certain drawbacks including the need for contrast agents and exposure to ionizing radiation (in the case of CT and venography), which is why they are typically reserved for second-line techniques after US in the evaluation of superficial lesions [14].
Consequently, we have to slightly rearrange the order of some paragraphs in the Discussion to ensure continuity and coherence.
We sincerely hope that we have correctly understood and implemented your suggestions. In any case, we remain fully available to make any further revisions that may be required.